# Impact of Climate Change Beliefs on Youths’ Engagement in Energy-Conservation Behavior: The Mediating Mechanism of Environmental Concerns

**DOI:** 10.3390/ijerph19127222

**Published:** 2022-06-13

**Authors:** Ping Han, Zepeng Tong, Yan Sun, Xuefeng Chen

**Affiliations:** 1Key Laboratory of Behavioral Science, Institute of Psychology, Chinese Academy of Sciences, Beijing 100101, China; gao45402@126.com (P.H.); tongzp@psych.ac.cn (Z.T.); suny@psych.ac.cn (Y.S.); 2Department of Psychology, University of Chinese Academy of Sciences, Beijing 100049, China

**Keywords:** carbon neutrality, climate change, environmental concern, energy-conservation behavior, adolescent

## Abstract

Global climate change presents a profound threat to the survival and continued development of humanity. The present study featured a survey of 3005 adolescents in China on 13 December 2021, aiming to determine whether climate change beliefs (including recognition of the existence of climate change, cognition of the causes of climate change, and climate change risk perception) influence their engagement in energy-conservation behaviors. Concurrently, the psychological mechanism underlying the influence of environmental concerns on the above relationship was also tested. The results showed that, among youths, climate change belief positively predicts engagement in energy-conservation behaviors. Specifically, awareness of the existence of climate change, knowledge of the causes of climate change, and climate change risk perception all positively predict engagement in energy-conservation behaviors. Further, environmental concerns were found to play a mediating role in the relationship between climate change beliefs and energy-conservation actions. From a practical perspective, the government and education departments should guide young people to develop accurate perceptions of climate change, and should raise their awareness of energy conservation and social responsibility, which should lead to their development of energy-conservation habits.

## 1. Introduction

Climate change has become one of the biggest challenges to human society in the 21st century [1], and has produced a series of direct or indirect impacts on human beings [2]. There is a growing demand for the world to achieve “net-zero CO_2_ emissions” by 2050; this goal is in the interests of all mankind and, as a result of the potentially serious consequences of high CO_2_ emissions, must be achieved as soon as possible. In the international energy market in 2021, the global energy crisis marked by the continuous rise in the prices of natural gas, coal, and oil would coexist with the climate problem and the energy transition boom [3]. Issues such as global climate change, local environmental governance, and regional sustainable development all relate to the matter of energy. At present, about 81% of the world’s primary energy comes from fossil fuels, of which oil accounts for 31.9%, coal for 27.1%, and natural gas for 22.1% [4]. In recent years, a shift in energy production and consumption mechanism has been promulgated as a vision of “sustainable development”. China is the world’s largest energy producer and consumer country. The energy structure of our country is dominated by coal, with supplementation from multiple energy resources [5]. As the world’s largest energy consumer, China accounted for about 22% of global energy consumption in 2018. Efforts to reduce energy consumption and improve energy efficiency from China are crucial to the global energy structure and sustainable development [6]. Energy efficiency refers to the ratio of useful output to energy input in a production process [7]. The role of energy efficiency is to reduce environmental pressure [8]. According to the China Energy Statistical Yearbook 2019, judging by energy consumption per unit of GDP, China’s energy efficiency level in economy is still relatively low [7].

China currently has the highest rate of energy waste in the world; at present, approximately 400 million metric tons of standard coal are wasted annually in China, accounting for 12–16% of the country’s energy production [9]. The entire process, from energy extraction to terminal consumption, contains waste [9].

With the rapid development of urbanization, the significant growth of residential energy consumption in China has become the main contributor to energy consumption. The use of energy resources has the following traits: abundant users, significant autonomy, and serious energy waste [10]. Since most residential energy consumption takes place in personal residence, accounting for 10.6% [11] of China’s total energy consumption, household energy consumption has become the second largest contributing factor (less only than the industrial sector) to the growth in China’s energy consumption and carbon emissions [12]. According to Chen [13], Chinese households emit, on average, almost 2.7 metric tons of carbon per capita annually; overall, 21% of the total carbon emissions of the entire Chinese society is associated with household energy consumption [13]. Urban residents’ daily activities, such as cooking, heating, lighting, and transportation, generally involve consumption of energy in the form of electricity and water supply; further, the products and services used to perform these activities emit CO_2_ during their entire cycle of development, production, and recycling. Notably, the carbon emissions resulting from such consumption increase as the standard of living rises, imposing an increasingly negative impact on the ecological environment [14]. Thus, energy conservation and carbon reduction at the consumer end can determine the total amount of carbon emissions [15].

In recent years, the Chinese government has adopted the development strategy of “expanding domestic demand to drive economic growth”; however, the household consumption pattern is characterized by resource consumption and heavy pollution [16]. Wang et al. [17] suggested that urban residents’ daily consumption behavior, by generating CO_2_ and other greenhouse gases, is the primary contributor to the total carbon emissions of the entire society. Thus, it is important to encourage the public to adopt a low- or zero-carbon lifestyle. Net-zero CO_2_ emissions into the atmosphere is referred to as “carbon neutrality”, and achieving this requires the capturing, utilization, and/or storage (through forest carbon sinks, artificial technologies, or engineering means) of the CO_2_ emissions produced during human activities [16]. For example, frameworks for achieving carbon-peaking and carbon-neutrality goals have suggested that governments should foster, among the public, full comprehension of the meaning of low-carbon consumption, promote adherence to the principle of prioritizing energy conservation, encourage recycling of resources, and implement multi-level and multi-form education that can engage more people in efforts to achieve carbon-emission reductions and promote zero-carbon behavior [16]. The ultimate goal is to form a social consensus on low-carbon and green development, which should facilitate the wider adoption of such behaviors in communities in the future.

There are various strategies for adjusting energy-consumption structures, improving energy efficiency and, ultimately, achieving emission-reduction targets. In particular, efforts designed to promote societies’ adaption to and mitigation of climate change should be focused on the younger generation [18]; their beliefs and actions concerning climate change will determine the future ecological environment of our country. In particular, early adolescence is a critical period for developing understanding of the issue of climate change [18]. Studies have shown that young people in China (overall, 17% of China’s population is aged under 15 [19]) lack basic knowledge of climate change, which leads to an inability to correctly and objectively understand the issue [20].

Pro-environmental behaviors (turning off lights, recycling, etc.) are daily actions that can be beneficial for the environment. However, pro-environmental behaviors cannot be merely individual, sporadic actions; they must be implemented in daily life. For example, reducing daily water and electricity use in order to lower everyday energy consumption [21]. This also means that the formation of pro-environmental behaviors does not occur instantly; consumption habits and behavioral patterns are cultivated over a long period of time. Many factors, including personal factors and social factors, can influence people’s engagement in pro-environmental behaviors; these can include one’s childhood experiences, knowledge, personality traits, self-efficacy, worldview, and values [22]. Adolescents are a high energy-consumption group, and will become adult consumers in the future. Their beliefs regarding climate change will inevitably affect their engagement in pro-environmental behaviors. Thus, knowledge of the current low-carbon, energy-conservation behaviors of the younger generation is crucial for addressing climate change.

Some studies have suggested that environmental concern is an effective indicator of pro-environmental behavior, as consumers with higher levels of environmental concern are more inclined to buy green products [23]. Othman [24] found that exposure to news reports on environmental issues and personal experience of such issues can increase one’s level of environmental concern, thereby promoting willingness to engage in environmental behaviors. Similarly, Vicente-Molina et al. [25] discovered that knowledge of environmental issues has a significant positive impact on individuals’ engagement in pro-environmental behaviors. Further, Tanner and Kast, applying the New Ecological Paradigm (NEP) scale, demonstrated the significant promoting effect of environmental beliefs on environmental behavior. Meanwhile, many scholars believe that age is a determining variable for environmental concerns; young people generally care more about the environment when compared to older adults [26]. However, domestic research has provided conflicting results in this regard. Shen et al. [27] conducted a survey in Shanghai and reported that age is positively correlated with environmental concerns; in contrast, Hong and Fan [28] and Hong et al. [29] have argued the opposite regarding the variable of age.

Previous studies have failed to clarify whether climate change beliefs can predict energy-conservation behavior, but have shown that middle school is a critical period for one’s adaptation to the world and development of a sense of ability to transform the world [30]. The present study seeks to examine the impact of climate change beliefs on adolescents’ energy-conservation behaviors and, to achieve this, investigates a sample of middle-school students. This research also aims to verify the psychological mechanism by which environmental concerns influence adolescents’ climate change beliefs and energy-conservation behaviors. Such investigation could inform the development of education and guidance for improving youths’ engagement in pro-environmental behaviors, thereby fulfilling one of the basic prerequisites for achieving carbon neutrality.

### 1.1. Energy-Conservation Behavior

Energy is an indispensable force for sustainable social development. In China, energy conservation is important for improving energy efficiency, regulating and improving the country’s energy structure, and promoting the development of low-carbon economy among the energy industry [31]. In 2011, 32.2 billion metric tons of carbon were emitted worldwide as a result of the use of fossil fuels to power human economic activities [32]. Over the past 40 years, energy consumption per unit of gross domestic product (GDP) has fallen by over 4% annually on average, with a cumulative drop of almost 84% [32]; this represents a remarkable achievement in energy conservation and consumption reduction. However, when compared with statistics for other countries, China’s energy consumption per unit of GDP is still 1.5 times the world average [32]. A country’s level of carbon emissions is closely related to its urban residents’ cognition of and willingness to implement energy-conservation behaviors [14].

The Ministry of Science and Technology’s research on the “Quantitative Indicators of National Energy Conservation and Emission Reduction Potential” indicated that alterations to urban residents’ daily life habits have huge energy-saving and emission-reduction potential. Interventions on energy conservation can reduce household energy consumption by 10–30%. Furthermore, behavior-driven energy conservation strategies require less capital and time investment than other approaches [33]. If everyone actively participates in such efforts, the total energy saving of Chinese residents will be approximately 77 million metric tons of coal equivalent per year, corresponding to a reduction of 200 million metric tons in CO_2_ emissions. Some studies have reported that low-carbon policies are widely regarded as an effective institutional arrangement for encouraging urban residents to adopt energy-conservation behaviors [34]. Thus, the construction of a low-carbon culture should be emphasized and popularized. In the existing literature, studies on the factors associated with energy-conservation behaviors have mainly focused on two aspects: personal situation and the external environment. For instance, regarding individual objective factors, level of individual income [35], age [36], and education [37] have been found to impact willingness to perform energy-conservation behaviors. The Big Five personality traits have different effects on energy-conservation behaviors among residents. There is a correlation between personality traits and household energy-conservation behaviors [38]. Meanwhile, several studies have reported subjective influences on urban residents’ engagement in energy-conservation behaviors. For example, Lu [36] suggested that residents’ awareness of energy conservation has an important impact on energy consumption in buildings. Meanwhile, Yang et al. [39] found a correlation between residents’ awareness of energy conservation and their carbon emissions, including their use of energy-saving products, saving of water and electricity, and their transportation choices. Intention is an important influencing factor of behavior [40]. It reflects an individual’s desire of attempts [40]. If a person is considered to have a strong desire to conserve energy, he/she will exhibit energy-conservation behaviors. Studies have confirmed that residents’ level of ecological awareness is a predictor of their engagement in energy-conservation behavior [41]. Age and electricity expenditure are negatively correlated with engagement in energy-conservation behaviors, while one’s perceptions of environmental and energy pressures promotes engagement in such behaviors [42]. Thus, a feasible strategy for promoting household energy conservation is expanding the dissemination of relevant information and shifting cognitive value norms among residents [43]. Further, enhancing urban residents’ engagement in energy-conservation behaviors would also be a good means of implementing energy-efficiency strategies.

In this survey, which considers the energy-conservation behaviors of adolescents, we mainly consider two types of daily, pro-environment activities: water and electricity saving.

### 1.2. Climate Change Beliefs

The United Nations Framework Convention on Climate Change defines climate change as “a change of climate which is attributed directly or indirectly to human activity that alters the composition of the global atmosphere and which is in addition to natural climate variability observed over comparable time periods” [44]. Thus, addressing climate change is an issue for mankind as a whole. Studies have shown that climate change beliefs are the most significant explanatory variable for determining engagement in environmental-protection behavior [45]. The public’s cognition of climate change and other environmental problems mainly comprises concern regarding the implications of such problems and understanding of the problems’ causes and harms [46]. Thus, the present study discusses climate change beliefs among teenagers from three aspects: the existence of climate change, causes of climate change, and perceived risk of climate change.

#### 1.2.1. Existence of Climate Change

Studies have shown that beliefs concerning climate change can influence individuals’ opinions on the issue and one’s support for corresponding mitigation policies [47]. Climate change impacts everyone’s daily life. Hong and Fan [28] compared public perception and action on climate change across 31 countries, and concluded that, when compared with developed countries, public awareness of climate change in China is relatively low; concurrently, adoption of pro-environmental behaviors among citizens is also limited. Xie and Chen [48] proved that the public’s concern regarding, and perception of, climate change significantly affects their willingness to act. Thus, examining public perception of climate change can reveal the extent to which the issue is recognized by the public and is influencing people’s actions. Research on public acceptance of climate change is necessary to determine the public’s perceptions of the nature of the issue, predict their responses, and identify mitigating measures that can be suitably introduced in society. Thus, the extent to which young people in China understand the facts concerning climate change deserves attention. It is reasonable to assume that adolescents who believe in climate change will adopt energy-conservation behaviors.

#### 1.2.2. Causes of Climate Change

Studies have shown that an accurate understanding of the causes of climate change is an essential determinant of one’s adoption of environmental actions and support for climate-protection policy measures [49]. Human activities have been shown to be causing changes in components of the climate, including in regional climate and the frequency of extreme events [50]; however, the Chinese public has only a limited understanding of the causes and impacts of climate change [28]. Relatively few Chinese people have a scientific understanding of the causes of climate change, and many do not understand that greenhouse gas produced by human activities is the main cause of climate change [51]. School education is the largest contributor to teenagers’ perceptions of climate change. In 2017, the Chinese curriculum standards for the subject of science were revised, but detailed teaching guidelines for respective age groups are still lacking [52]; climate change is briefly discussed in high-school geography, but using a top-down educational approach [53]. Notably, teenagers’ perceptions of the causes of climate change, as well as whether such understanding affects their low-carbon energy-consumption behavior, is yet to be studied.

#### 1.2.3. Climate Change Risk Perception

O’Connor et al. [54] stated that climate change risk perception refers to one’s understanding of the negative consequences of global warming for individuals and society as a whole. Zhang et al. [55] defined the term as objective perception and subjective feeling regarding the risk of climate change to daily life. Specifically, the emergence of risk is objective, but the generation of risk perception is subjective. This study adopts O’Connor et al.’s above definition. Studies have shown that over 70% of the Chinese population shares a high degree of concern about climate change due to its worrying adverse effects [56].

Kim and Hwang [57] found that the perceived severity of climate change is an important indicator that positively correlates with one’s willingness to adopt pro-environmental behaviors. Similarly, Masud et al. [58] concluded that awareness, knowledge, and risk perception of climate change significantly affect one’s attitude and pro-environmental behaviors. Moreover, Zhou and Tang [59] argued that media use can influence engagement in pro-environmental behaviors through the chain mediating effect of environmental knowledge and environmental risk perception. People are more likely to embrace pro-environmental behaviors when they fully understand the adverse effects of inaction; thus, climate change risk perception may directly or indirectly affect individuals’ engagement in pro-environmental behaviors.

In conclusion, this paper hypothesizes that, among adolescents, climate change beliefs are highly correlated with engagement in energy-conservation behaviors.

**Hypothesis** **1** **(H1).***Climate change beliefs positively predict young people’s engagement in energy-conservation behaviors*.

### 1.3. Environmental Concerns

Dunlap and Jones [60] defined environmental concern as the degree to which people are aware of environmental problems and support measures for resolving them, or individuals’ willingness of to make personal efforts to address these problems. Concurrently, Stern et al. [61] highlighted that environmental concern is related to individuals’ fundamental values, with the level of such concern being ultimately determined by their core values. Related empirical studies have also argued that environmental concern is positively correlated with altruistic and ecosphere values, while negatively correlated with egoistic values [62]. Values are integrated in verbal and nonverbal symbols, communication patterns, daily routines, material culture, social institutions, and the ways people structure and relate to their natural and social surroundings [63]. Values define and bind groups, organizations, and societies, serve an adaptive role, and are typically stable across generations. When abrupt value changes occur, they are in response to substantial alterations in the social–ecological context. Such changes build on prior value structures and do not result in complete replacement [64].

The objective reality of environmental deterioration attracts people’s attention; however, their subsequent actions depend on their view of social life and the environment [65]. The degree of environmental protection behavior an individual engages in is proportional to his/her level of environmental concern. The more people pay attention to environmental issues, the more likely they will act [66]. Studies based on rational behavior theory and value–belief–norm theory showed that public environmental concerns rooted in individuals’ consciousness, social culture, customs, etc., have a different impact on end-use energy-consumption behavior than informal mandatory constraints such as laws and regulations [67]. Environmental concerns lead to the gradual formation of benign personal behavior norms which drive the public to engage in environmentally friendly behaviors. Environmental concern formed on the basis of “consciousness-behavior” analysis is being increasingly recognized in research. While environmental concerns are determining to one’s engagement in energy conservation. Liobikiene et al. [68] suggested that awakening of environmental concerns can encourage the public to conserve energy, which, in turn, can lead to green consumer behaviors. After comparing a large number of related studies, Tsuda et al. [69] reported an interactive relationship between environmental concerns and engagement in energy-conservation behaviors. Similarly, Pothitou et al. [70] identified a significant positive correlation between environmental concern and engagement in pro-environmental behaviors.

Through the above literature review, it can be concluded that environmental concerns are the driving force of climate change beliefs and engagement in energy-conservation behaviors. Values are integrated in verbal and nonverbal symbols, communication patterns, daily routines, material culture, social institutions, and the ways people structure and relate to their natural and social surroundings [63]. Values define and bind groups, organizations, and societies, serve an adaptive role, and are typically stable across generations. When abrupt value changes occur, they are in response to substantial alterations in the social–ecological context. Such changes build on prior value structures and do not result in complete replacement [64]. However, the factors that determine the strength of environmental concerns have not yet been identified, and there is also a lack of in-depth research on the mechanism involved in the above relationship. Therefore, the present research intends to study how environmental concerns drive engagement in energy-conservation behaviors in the Chinese context. Specifically, this paper analyzes the influencing factors for environmental concerns and engagement in energy-conservation behaviors among young people in order to develop policy suggestions that can assist the construction of a low-carbon society. Additionally, this study adopts structural equation modeling to verify the mediating impact of environmental concerns on the relationship between climate change beliefs and engagement in energy-conservation behaviors among adolescents.

**Hypothesis** **2** **(H2).***Among adolescents, environmental concerns mediate the influence of climate change beliefs on energy-conservation behaviors*.

### 1.4. Current Study

In conclusion, the present study aims to investigate the influence of climate change beliefs on engagement in energy-conservation behaviors among adolescents, while also verifying the psychological mechanisms underlying environmental concerns. Climate change beliefs contain three dimensions, namely, the occurrence of climate change, attribution of climate change and perception of climate change risk. Hypothesis 1 (H1): Climate change beliefs positively predict young people’s engagement in energy-conservation behaviors; Hypothesis 2 (H2): Among adolescents, environmental concerns mediate the influence of climate change beliefs on energy-conservation behaviors. This paper’s findings may provide guidance for promoting energy-conservation behaviors among teenagers. Additionally, this research regards environmental concern as a mediating variable that can provide valuable guidance for environmental education for youths. The conceptual model for this paper is shown in Figure 1.

## 2. Materials and Methods

### 2.1. Participants

This study involved a survey of middle-school students in Beijing, China. The study was approved by the Ethics Committee of the Institute of Psychology, Chinese Academy of Sciences. Overall, 3005 questionnaires were distributed, and 2881 valid results are retrieved, giving a recovery ratio of 95.87%. Among the participants, 1432 were girls, 1408 were boys, and 41 did not report their sex. The age range was 8–15 years, with an average of 12.79 years.

### 2.2. Measures

#### 2.2.1. Climate Change Beliefs

All participants in this study read a definition of climate change that was based on environmental psychographics. Three items on the survey related to the respondents’ climate change beliefs. One concerned whether the respondents accepted the existence of climate change: “I think climate change is happening”; one concerned the causes of climate change: “If you think climate change is happening, what do you think is the main cause?”; and one concerned climate change risk perception: “How worried are you about climate change?” Respondents answered each of the above questions using an 11-point scale. Responses for the question of whether climate change exists were scored as follows: 0 = “strongly disagree”, 10 = “strongly agree”; higher scores represented stronger belief in climate change. The responses for the question concerning climate change causes were scored as follows: 0 = “mainly caused by natural causes”, 5 = “natural causes and human activities have an equal effect”, 10 = “mainly caused by human activities”; here, higher scores indicated more accurate knowledge regarding the causes of climate change. Finally, the responses for the question concerning climate change risk perception were scored as follows: 0 = “no worries at all”, 10 = “very concerned”; here, higher scores represented stronger perceived risk of climate change. Students gave their answers based on their own personal understandings.

#### 2.2.2. Environmental Concerns

The survey contained four items concerning environmental concerns: “Environmental problems will not have serious negative effects for human beings and nature”; “Protecting the environment is very important for our future”; “We must maintain environmental balance”; “I am not worried about environmental problems.” The students responded to each of the above items using an 11-point scale, which indicated their level of agreement with each statement (0 = “strongly disagree”, 10 = “strongly agree”). Higher scores for “Protecting the environment is very important for our future” and “We must maintain environmental balance” and lower scores for “Environmental problems will not have serious negative effects for human beings and nature” and “I am not worried about environmental problems” indicated stronger environmental concerns. The internal consistency of this environmental-concerns scale was favorable; for this study, its Cronbach’s alpha value was 0.64.

#### 2.2.3. Energy-Consumption Behaviors

Based on consideration of common low-carbon consumption behaviors, we developed for this study an energy-conservation-behavior scale. This scale contained the following items: “I turn off the lights every time I leave a room”; “I unplug my phone or other appliances when they are fully charged in order to save electricity”; “I use air conditioners or heaters sparingly; for example, in summer the air conditioner is only used when the temperature is 26 °C or higher, and in winter, when heating is on, I keep windows closed to reduce unnecessary energy waste”; “I take measures to reduce water consumption when I take a bath (such as turning off the shower when applying shampoo).” The above items were scored using an 11-point scale, which the participants used to report how closely each statement accorded with their usual behaviors (0 = “strongly disagree”, 10 = “strongly agree”). For each respondent, the scores for all four items were summed and averaged, and higher scores indicated stronger willingness to engage in energy-conservation behaviors. For the present study, the Cronbach’s alpha coefficient for this scale was 0.88, indicating promising internal consistency.

### 2.3. Procedure

All participants were required to read carbon-neutrality-related information provided in the questionnaire before answering the questions. The questionnaire comprised five sections: carbon-neutrality cognition, carbon-neutrality education, carbon-neutrality behaviors, environmental psychological characteristics, and demographic information. The third and fourth sections are expected to provide key data for this research. In the behaviors section, we focused on the adolescents’ engagement in low-carbon, environmental-protection behaviors. In the psychology section, we focused on the mediating mechanism in the relationship between climate change beliefs and environmental concerns among teenagers. The final section concerned the measurement of demographic variables.

### 2.4. Data Analysis

IBM SPSS Statistics for Windows, version 26.0 (IBM Corp., Armonk, NY, USA), was used to perform our data analysis; this included the use of descriptive statistics and correlation coefficient analysis. To examine the mediating role of environmental concerns, we applied a stepwise regression approach.

## 3. Results

### 3.1. Impact of Climate Change Beliefs on Engagement in Low-Carbon, Energy-Conservation Behaviors

The results of this study provide preliminary evidence for the positive reinforcement effect of climate change belief on energy-conservation behavior of Chinese adolescents. The survey included 1406 boys and 1431 girls, and there was no significant difference between the two groups in energy-conservation behavior. For boys, M = 5.89 ± 1.65, while for girls, M = 5.67 ± 1.38, showing a significant difference (*p* < 0.001).

Pearson correlation analysis was conducted on the three variables of climate change beliefs, environmental concerns, and engagement in energy-conservation behaviors. The results showed that the three variables of climate change beliefs positively correlated with engagement in energy-conservation behaviors and environmental concerns with a high level of significance. Engagement in energy-conservation behaviors was shown to be significantly positively correlated with environmental concerns, validating H1 (Table 1).

### 3.2. Mediating Mechanism of Environmental Concerns

We applied Pearson correlation analysis to test the mediating effect of environmental concerns on the relationship between climate change beliefs and engagement in energy-conservation behaviors. The results showed that the influence of climate change beliefs on young people’s engagement in energy-conservation behaviors occurs through both a direct effect and through a mediating effect of environmental concerns. Specifically, the direct effect of climate change beliefs on youths’ engagement in energy-conservation behaviors was significant (β = 0.241; 95% confidence interval [0.208, 0.275]), while the mediating effect of environmental concerns was also significant (β = 0.02; 95% confidence interval [0.012, 0.029]). The direct effect of climate change causes on youths’ engagement in energy-conservation behaviors was also significant (β = 0.122; 95% confidence interval [0.089, 0.155]), as was the mediating effect of environmental concerns (β = 0.026; 95% confidence interval [0.017, 0.037]). The direct effect of climate change risk perception on youths’ engagement in energy-conservation behaviors was significant (β = 0.274, 95% confidence interval [0.244, 0.304]), while the mediating effect of environmental concerns was also significant (β = 0.015, 95% confidence interval [0.008, 0.022]). The above results verify H2 (Figure 2, Figure 3, Figure 4 and Table 2).

School education is the most influential factor on perceptions of climate change among Chinese teenagers. The Curriculum Standards for Compulsory Education (2022 edition) present many changes, among which the objectives and content structure for Chinese students have been improved. The comprehensiveness and practicality of the course are improved by the focus on development of core literacy, selected and designed course content, and “interdisciplinary” learning activities. The new standards strive to go beyond disciplines and solve practical problems. Under the current mechanism of education in China, climate change will undoubtedly be introduced into the classroom. Climate change belief among teenagers will positively affect energy-conservation behavior.

## 4. Discussion

The data analysis conducted in the present research showed that, among Chinese youths, climate change beliefs positively predict engagement in energy-conservation behaviors, and that environmental concerns play a mediating role in this relationship. These research findings may provide a basis for the government and education departments to, among youths, promote the formation of correct understanding of climate change, enhance awareness of the need for energy conservation and sense of social responsibility, and develop energy-conservation habits.

### 4.1. Climate Change Beliefs Promote Energy-Conservation Behaviors among Teenagers

There is a large and emerging body of evidence concerning the social realities associated with climate change. The effectiveness of a country’s climate change adaptation and mitigation policies depends on the support these policies receive from the public. However, the public’s climate change beliefs are shaped by their ideology and knowledge of the issue [71]. It is especially important to understand the factors that may influence climate change beliefs among youths.

Research has shown that climate change beliefs are directly related to one’s experiences, and can influence one’s concerns about climate change [72]. Climate change beliefs are also the most significant explanatory variable determining one’s engagement in environmental-protection behaviors. Further, environmental beliefs also have a significant promoting effect on engagement in environmental behaviors [73]. Additionally, the public’s knowledge and risk perceptions of climate change have a strong influence on governments’ formulation and implementation of climate change mitigation policies [74] (knowledge and the quality of environment have obvious effects on climate change beliefs; however, environmental quality has greater explanatory power for climate change beliefs than climate change knowledge [45]). In this study, three variables concerning adolescents’ climate change beliefs (existence, causes, and risk perception of climate change) were found to be strongly correlated with energy-conservation behavior, indicating that Chinese adolescents have an accurate understanding of climate change and can consciously adopt low-carbon behaviors in their daily lives. Guiding young people to adopt energy-conservation behaviors in life and on campus can play a vital role in energy conservation and emission reduction.

### 4.2. Mediating Effects of Environmental Concerns

The results of the present study show that environmental concerns are closely related to climate change beliefs (the existence, causes, and risk perception of climate change), and youths’ engagement in energy-conservation behaviors. Meanwhile, behavioral theorists Oreg and Katz-Gerro [75] argued that environmental concerns and other variables combine to motivate one’s environmental behaviors, and ultimately lead to engagement in environmental-friendly behaviors.

### 4.3. Implications for Management Practice

The results of this study have several implications for educating adolescents on energy-conservation behaviors. First, the intensity and coverage of the publicization of climate change-related knowledge are insufficient. Although China has taken many measures to publicize and provide guidance regarding the need to adopt low-carbon behaviors (e.g., through creating low-carbon-consumption days, low-carbon community building), the target population is not broad enough, especially among the younger age groups. Popularization of knowledge concerning the need to reduce carbon consumption and adopt a low-carbon lifestyle is insufficient. Further, popularization of professional knowledge such as carbon footprint, carbon credit, carbon trading, and the cost-effectiveness of low-carbon products also requires further promotion. Strengthening the publicity of and education on climate change in middle school can help young people understand climate change, learn its causes, and correctly perceive the associated risks, allowing them to consciously practice energy-conservation behaviors in their daily lives. Second, education is the second largest energy-consuming division in the service sector. Positive environmental awareness, knowledge of renewable energy, and active energy-conservation behaviors of the principal are related to school environmental energy conservation actions and energy upgrades. Educational supervision of principals should be strengthened [76]. Climate change education for teachers and parents is also important. Environmental education courses related to climate change should be held regularly, while class tours to Meteorological Science Exhibition Halls should also be arranged. In particular, the awareness of teachers and parents should be increased so that they can educate, influence, and guide young people’s adoption of low-carbon, environmental-protection behaviors. Third, various energy-conservation measures that encourage adolescents’ adherence to low-carbon behaviors should be adopted. These measures could include energy-saving-knowledge competitions and the provision of moral incentives, such as voluntary posts on campuses that publicize low-carbon, environmental-protection measures.

### 4.4. Limitations and Prospects

This study contains some limitations. First, data for all variables in this study were self-reported, creating a risk of social-desirability bias. As the questionnaires were first distributed to the respondents’ teachers, respondents’ desire to obtain praise, manage their self-images, publicize their personal values, etc., may have influenced the actual results. Second, the current research is cross-sectional; without longitudinal data on the participants’ climate change beliefs, it is difficult to infer whether adolescents’ climate change beliefs can be influenced by social circumstances.

Future research should focus on individuals at different school levels and in different regions. Stratified sampling can improve the overall reliability and validity of the results, and help make the research more in-depth and comprehensive. Second, future research may consider controlling for respondent tendencies, induced by the questionnaire method, to report engagement in pro-environmental behaviors. Other experimental approaches could be adopted to make participants less aware of the purpose of the survey, which could help to capture their true attitudes towards energy-conservation behaviors and willingness to engage in such behaviors. Additionally, as the questionnaire method cannot clarify temporal continuity of beliefs, future research should conduct continuous observations at different time points. Third, since the average age of the tested group is about 13 years old, they are in the early stage of physical and psychological development. Their relative physical maturity and lag in psychological development contribute to a coexistence of semi-mature and semi-naive. At the same time, due to lack of practice, junior high students are easily affected by the surrounding environment, including society, school, parents, peer groups, etc. Their value orientations are diversified and complicated [77].

## 5. Conclusions

The results of this study provide preliminary evidence for the positive reinforcement effect of climate change beliefs on engagement in energy-conservation behaviors among youths, and are expected to provide a reference for publicity of and education regarding energy-conservation behaviors. From the perspective of psychological science, climate change beliefs have a unique and universal psychological effect that can help young people strengthen their awareness of their low-carbon responsibility, improve their knowledge of climate change, develop a low-carbon consumption habit, and consciously practice energy-conservation behaviors. Climate change beliefs among adolescents have an important influence on their formation of environmental concerns; the stronger an individual’s climate change belief, the higher his/her environmental concern. To be more specific, teachers can carry out campus ecological education by introducing the knowledge of climate change and its possible results, pointing out practical energy conservation measures for young people, formulating guidelines for energy saving and emission reduction in order to put subjective environmental concern into practice.

## Figures and Tables

**Figure 1 ijerph-19-07222-f001:**
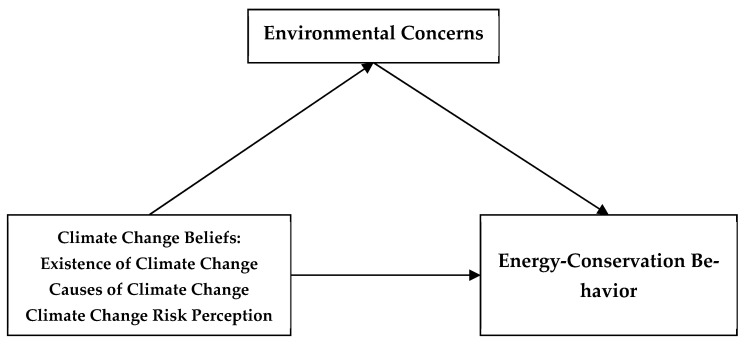
Schematic diagram of the theoretical model.

**Figure 2 ijerph-19-07222-f002:**
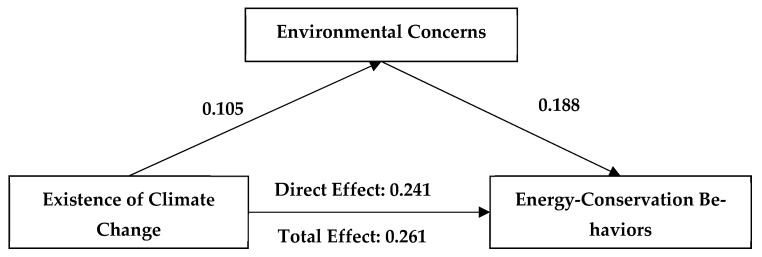
A mediating model for the effects of environmental concerns and acceptance of the existence of climate change on engagement in energy-conservation behaviors. All coefficients unstandardized.

**Figure 3 ijerph-19-07222-f003:**
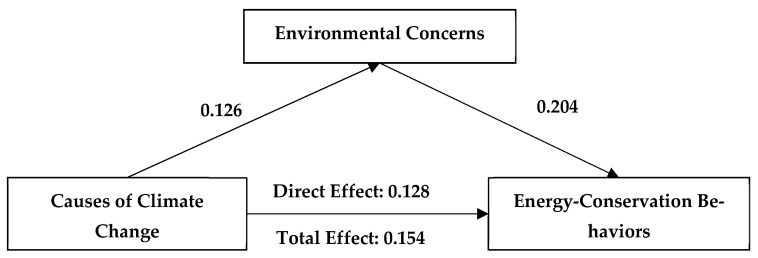
A mediating model for the effects of environmental concerns and knowledge of the causes of climate change on engagement in energy-conservation behaviors. All coefficients unstandardized.

**Figure 4 ijerph-19-07222-f004:**
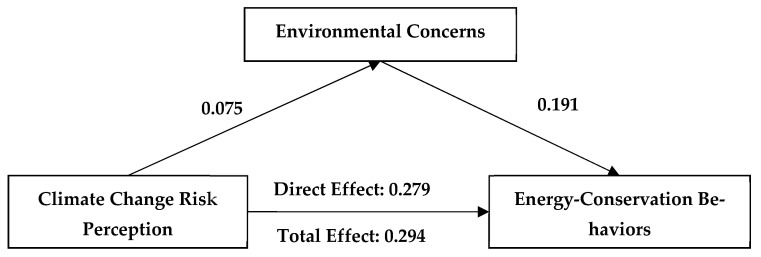
A mediating model of the effects of environmental concerns and climate change risk perception on engagement in energy-conservation behaviors. All coefficients unstandardized.

**Table 1 ijerph-19-07222-t001:** Means, standard deviations, and correlation coefficients for the study variables.

	Mean Value	Standard Deviation	Energy-Conservation Behavior	Environmental Concerns	Existence	Causes	Risk Perception
Energy-conservation behavior	8.074	2.142	1				
Environmental concerns	5.783	1.525	0.172 **	1			
Existence	8.472	2.184	0.266 **	0.151 **	1		
Causes	7.389	2.285	0.164 **	0.190 **	0.377 **	1	
Risk perception	7.662	2.410	0.330 **	0.119 **	0.428 **	0.394 **	1

** At the 0.01 level (two-tailed), the correlation is significant.

**Table 2 ijerph-19-07222-t002:** Total effects, direct effects, and mediating effects.

Energy-Conservation Behavior (Y)	β	95% Confidence Interval	Proportion (%)
**Existence**			
Total effect	0.261	[0.226, 0.295]	
Direct effect	0.241	[0.206, 0.275]	92.33
Mediating effects of environmental concerns	0.02	[0.012, 0.029]	7.67
**Causes**			
Total effect	0.155	[0.120, 0.188]	
Direct effect	0.128	[0.094, 0.162]	82.58
Mediating effects of environmental concerns	0.027	[0.017, 0.390]	17.42
**Risk Perception**			
Total effect	0.294	[0.262, 0.324]	
Direct effect	0.279	[0.248, 0.309]	94.89
Mediating effects of environmental concerns	0.016	[0.008, 0.025]	5.44

## Data Availability

All data were uploaded on the Figshare. Other researchers can download the dataset at https://figshare.com/s/be2326bff656809899d7 (accessed on 10 June 2022).

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
