# Peer review of "Impact of Climate Change Beliefs on Youths’ Engagement in Energy-Conservation Behavior: The Mediating Mechanism of Environmental Concerns"

_ijerph, 2022, doi:10.3390/ijerph19127222_

Round 1
Reviewer 1 Report
I am impressed not only by the reliable method of conducting the research but also by its professional and clear presentation.
I have only one comment. I don't think 2022 can be considered the mid-21st century. (line 30)
Author Response
Dear Reviewer,
Thank you so much for providing us with an opportunity to revise our manuscript. The following notes detail a re-write that has dealt with your comments. Each comment is presented, followed by the response given to it. Please see the attachment.
We are glad to continue to modify and improve our manuscript if you still have
any further questions.
Best,
Xuefeng Chen Professor
Key Laboratory of Behavioral Science
Institute of Psychology, Chinese Academy of Sciences
4A Datun Road, Chaoyang District, Beijing (100101), P.R. China

Reviewer 2 Report
Impact of Climate Change Beliefs on youths’ engagement in energy conservation behavior: the mediating mechanism of environmental concerns.
The paper contains some nice delivery and organization of the hypothesis and the Chronbach alpha testing. However, the paper contains some statements and assumptions that are not well supported by the manuscript or seem a bit extraneous/not directly on topic. These statements are located in the abstract, introduction, discussion, and conclusion. Further, the correlation findings do not appear to be properly interpreted. Examples follow.
1. Some matters regarding energy efficiency and consumption need clarification or a sound and consistent grounding. This section includes several citations, but not a clear depiction of energy flow (input, output, and efficiency). For instance, energy use in the United States is broken into categories: electricity, space heating, transportation, and high-intensity heat for industrial processing. We measure net energy for the sake of efficiency. There is a huge loss of energy (e.g., 80%) in our fossil fuel-based energy systems. And, electricity generation is typically generated by power plants, not homes. With this in mind, a home receiving electricity from a fossil fuel feed cogeneration plant will have different energy efficiency, and even that efficiency will be based on the type of fuel (coal, oil, or natural gas). So to say 21% of all carbon emissions are associated with households, and that in essence pro-social behavior begins at home contains some major disconnects in describing the energy system and the agency and type of knowledge people may need to conserve energy and reduce carbon emissions. (As a caricature, students will focus on plastic bottles and straws rather than more direct efforts. (Students need to understand the systems). Another factor to clarify is the impact of the consumption of goods by people inside and outside the home. In the United States, we use the IPAT or ecological footprint model. Or the country’s data is often divided per capita. This manuscript could use some clarification as to how the data was generated, without the reader needing to verify the data and its consistency across statements through each cited report.
2. Following clarification to the introduction, I suggest that the paper succinctly identifies the problem that you are trying to tackle with this work. (For instance, have a clear research question to guide the outline.)
Then present your variables and your hypothesis. Then stick to your hypothesis and don’t make other statements of findings that you have not included in this study, such as statements made in lines 398-400; and 437- 443.
3. Ideologies and personal beliefs are not necessarily fungible (i.e. easily changed). See Manfredo, M. J., J. T. Bruskotter, T. L. Teel, D. Fulton, S. H. Schwartz, R. Arlinghaus, S. Oishi, A. K Uskul, K. Redford, S.Kitayama, and L. Sullivan. 2017. Why social values cannot be changed for the sake of conservation. Conservation Biology 31: 772–780. There are also cultural elements to belief systems (see work by Kitayama cited above).
4. The interpretation of correlation coefficients is not according to my knowledge or reference materials. It appears that this manuscript is using the confidence level of the coefficient, not the actual coefficient. Coefficient values (r) are interpreted as: r > 0.7 to 1.0 = strong effect; r < 0.3 are weak. Those values between weak and strong are considered moderate. This will change your interpretation of your results.
Author Response
Dear Reviewer,
Thank you so much for providing us with an opportunity to revise our manuscript. The following notes detail a re-write that has dealt with comments of your comments. Each comment is presented, followed by the response given to it. Please see the attachment.
We are glad to continue to modify and improve our manuscript if you still have any further questions.
Best,
Xuefeng Chen, Professor
Key Laboratory of Behavioral Science
Institute of Psychology, Chinese Academy of Sciences
4A Datun Road, Chaoyang District, Beijing (100101), P.R. China

Reviewer 3 Report
The manuscript deals with an interesting scientific topic, having an insightful research objective, a robust methodology and insightful remarks. Indeed, the research objective “to study how environmental concerns drive engagement in energy-conservation behaviors in the Chinese context”, it has been approached in a creative and systematic manner. However, there is still room for organizational, narrative, and argumentation improvements, prior it to be accepted for publication at the “International Journal of Environmental Research and Public Health”. To this end the following review comments can be considered.
1) The theoretical coverage can be enhanced by more and relevant published papers. Therefore, the recommended list of published papers can be considered and added at the revised manuscript, offering a fresher and pluralistic international overview in the fields of energy conservation, climate change, and low carbon measures/behaviours, accordingly.
2) The repetition of the phrase of “present study” aims, seeks, …..should be focused on one or two text extracts, avoiding the dispersed notation of its research objective through the narrative flow.
3) The exact period, calendar year and season, in which the survey was conducted, it can be denoted in the relevant Abstract section.
4) The verb tenses for past and completed actions (either by the authors’ own analysis and for the citations recalled) they are either past, or, passive voice. Therefore, the present verb tense for already completed actions in the past they should be avoided.
5) A research limitation is that the first four subsections of main section 1 contain only theoretical development of the literature overview, being overly deprived by quantitative information that is so much vital for authors to better and rigorously approach all citations recalled. Therefore, a portion of the theoretical coverage can be better reorganized considering:
-a chronological overview of the last three decades in the Chinese context, and/or
-a geographical overview of a) international context and b) Chinese context, and/or
-a themes overview by grouping all literature production in themes of analysis in the Chinese context, like: energy context, climate change-environmental context, and socio-behavioural context.
-accompany the entities of Figure 1 with a Table in which at the left side column these terms will be recalled “as is”, while at another right-side column each one of these terms could be accompanied by one or more citations’ referred to.
Such an reorganization could upgrade the whole analysis and text content.
6) Another research limitation is that the, indeed impressive, sample surveyed is young students of about 13 years old. However, at the developmental stage of personality building there is not fully developed the self-consciousness, the self-identity, and the awareness in environmental, energy, or pollution issues. Of course students are the vital and dynamic part of any society but I am not convinced that such a survey could make credible and trustworthy conclusions; no because it is false, but the sample selection seems unsuitable for such informed and mature responses to be given and treated from the planners-authors-researchers. Students’ perceptions at such an age group should be considered unstable and “transitional”, being stabilized and instilled at later phases of their personality development, at least at 16 years old, where more socio-environmental “signals” from their surrounding world could shape their behaviour, either as consumers themselves of energy-demanding goods and products, or living at families with the other aged members: grandparents, parents, siblings, are working in agricultural, environmental-, energy-, services- sectors. Such critique should be developed at a distinct subsection 4.4, at the end of main section 4. Discussion. At this new subsection 4.4 authors are also recommended to make explicit conclusions of generalized applicability and usefulness, beyond the Chinese context, but certainly based on the behavioural and attributable relationships reported among the specific “Chinese youths” where “climate change beliefs positively predict engagement in energy-conservation behaviors, and that environmental concerns play a mediating role”.
7) At all sections 2, and 3, as well as at subsection 4.3, the relevant text extracts have to be validated and verified by cross-citations, where missing and where matching. Again for these text points the list of recommended publications can be considered and cited. Another reason of literature enrichment and refresh is that a main portion of the existing literature production was focused on rather dating citations, being published one decade ago or even earlier, thus, an updated literature production should better reflect the today behaviourism on the topics of environment, climate change, and energy-conservation perceptions, either at the Chinese society or abroad.
Scopus
EXPORT DATE:21 May 2022
Alomari, M.M., EL-Kanj, H., Topal, A., Alshdaifat, N.I.
26031239300;57215535881;57202337104;57195776604;
Energy conservation behavior of university occupants in Kuwait: A multigroup analysis
(2022) Sustainable Energy Technologies and Assessments, 52, art. no. 102198, .
https://www.scopus.com/inward/record.uri?eid=2-s2.0-85127767536&doi=10.1016%2fj.seta.2022.102198&partnerID=40&md5=dd78ec4ac28405ceeebf6a4a3df12ff4
DOI: 10.1016/j.seta.2022.102198
AFFILIATIONS: Electrical Engineering Department, Australian University (AU), Mishref40005, Kuwait;
Business Department, Nigde Omer Halisdemir University, Nigde, 51240, Turkey;
JoVision, Hamburg, 22083, Germany
ABSTRACT: The continuous increase in energy consumption has led to strategies worldwide to encourage energy conservation behavioral strategies to mitigate high demand and environmental problems. Limited literature focuses on the university occupants' energy conservation behaviors (ECBs). This study examined how three university occupant groups (faculty, students, and staff) performed in the Theory of Planned Behavior (TBP) hypotheses. The current study also aims to examine how the occupants differ in the dimensions of TBP affecting their ECBs. Model testing was done using partial least square structural equation modeling (PLS-SEM) and multigroup analysis (MGA). At the overall model, results showed that all predictors of ECB are significant. All hypotheses were supported except for direct relationships between EK and ECB and AWC and ECB. After applying the MGA, our research revealed that faculty demonstrated a more confident ECB and have a more positive INT than students and staff due to their EK and AWC. However, there is still a significant difference in the EK of occupants. Interestingly, this study found that students display greater trust in SNs than the levels of SNs reported by faculty and staff. This research suggests that universities should consider implementing unique energy education methods and approaches to each occupant. © 2022 Elsevier Ltd
AUTHOR KEYWORDS: Energy behavior; Energy conservation; Environmental knowledge; Multigroup analysis; PLS-SEM
DOCUMENT TYPE: Article
PUBLICATION STAGE: Final
SOURCE: Scopus
Ntanos, S., Kyriakopoulos, G.L., Anagnostopoulos, T., Xanthopoulos, T., Kytagias, C., Drosos, D.
57076831500;6603382498;14031080700;57217104674;25958192100;56394701400;
Investigating the Environmental and the Energy Saving Behavior among School Principals through Classification Algorithms
(2022) International Journal of Renewable Energy Development, 11 (2), pp. 449-461.
https://www.scopus.com/inward/record.uri?eid=2-s2.0-85128749782&doi=10.14710%2fijred.2022.43007&partnerID=40&md5=1aee3b0b5629da4aebde0d9433fffd99
DOI: 10.14710/ijred.2022.43007
AFFILIATIONS: School of Business, Economics and Social Sciences, Department of Business Administration, University of West Attica, 250 Thivon & P. Ralli str, Egaleo, 12244, Greece;
School of Electrical and Computer Engineering, Electric Power Division, Photometry Laboratory, National Technical University of Athens, Athens, 15780, Greece;
Department of Business Administration, DigiT.DSS.Lab, University of West Attica, 250 Thivon & P. Ralli str, Egaleo, 12244, Greece
ABSTRACT: Buildings are a significant energy consumption point since they account for 40% of the total energy demand and around 1/3 of greenhouse gas emissions. Energy-saving measures applied in the residential sector have led to a reduction in energy consumption during the last decade. On the contrary, such measures have not been widely applied in school buildings, although education is the second-largest energy consumer in the service sector. This paper aims to assess school principals' perceptions concerning energy saving and the environment since they are responsible for promoting energy-saving measures and investments and inspiring students and school personnel towards environmentally friendly behavior. Using survey data from Greek schools, we applied predictive classification models to locate the most critical variables that drive principals' perceptions of energy upgrading and energy-saving actions at school. Results revealed that the positive environmental perceptions of principals, the level of knowledge on Renewable Energy Sources (RES) and the active energy-saving behavior are related to energy-saving actions and energy upgrading in school environment. Furthermore, the creation of more RES oriented courses is related to positive energy-saving behavior and actions. Thus, emphasis should be put on educating and informing the school principals concerning RES technologies and energy-saving options since they are critical players in applying energy-saving measures in school buildings. © 2022. The Authors. Published by CBIORE.
AUTHOR KEYWORDS: classification algorithms; energy-saving; environmental behavior; renewable energy; school building; school environment
DOCUMENT TYPE: Article
PUBLICATION STAGE: Final
SOURCE: Scopus
Wang, J., Yi, F., Zhong, Z., Qiu, Z., Yu, B.
57202393887;57296202500;57296736700;57222655119;57207862819;
Diversity and causality of university students’ energy-conservation behavior: Evidence in hot summer and warm winter area of China
(2021) Journal of Cleaner Production, 326, art. no. 129352, . Cited 2 times.
https://www.scopus.com/inward/record.uri?eid=2-s2.0-85117132361&doi=10.1016%2fj.jclepro.2021.129352&partnerID=40&md5=be04da9f9f90947bd859f7d9110f2aa0
DOI: 10.1016/j.jclepro.2021.129352
AFFILIATIONS: College of Civil and Transportation Engineering, Sino-Australia Joint Research Center in BIM and Smart Construction, Shenzhen University, Shenzhen, China;
College of Civil and Transportation Engineering, Shenzhen University, Nanshan, Shenzhen, China
ABSTRACT: Energy conservation in universities is significant to sustainable development, and students' behaviors hold great potential for green university development. However, few studies focused on the diversity and causality of energy-conservation behaviors (ECBs) among university students, as well as behavior differences in social characteristics. In this study, university students' ECBs were defined, including habitual ECBs, restricted ECBs, and interpersonal ECBs through literature reviews. A theoretical model of behavior causality was constructed based on the refined theory of planned behavior. Four hundred and thirty-nine questionnaire data verified the hypothesized causality using a structural equation model. Diversities in social characteristics on behavior paths and determinants were explained through multi-group analysis and one-way analysis of variance. Results show that: (1) ECBs are all affected by comfort preference and perceived self-efficacy, and perceived self-efficacy is the most critical facilitator; (2) the determinants and their importance of ECBs vary with the behavior types; (3) the effect of comfort preference, energy-conservation value, and perceived self-efficacy acting on habitual ECBs are completely mediated by energy-conservation intention; (4) gender, grade, and discipline have no moderating effect on behavior paths, while they have significant differences in behaviors and factors, and it emphasizes the necessity to focus on males, higher grades, and social science students. This work generated fresh insight into the causality of students’ ECBs and differences of behaviors and determinants in social characteristics, which make contributions to the development of ECB-related theories. Besides, some theoretical references for university administrators were provided to develop energy management measures. © 2021 Elsevier Ltd
AUTHOR KEYWORDS: Energy-conservation behavior; Multi-group analysis; One-way analysis of variance; Structural equation model; University student
DOCUMENT TYPE: Article
PUBLICATION STAGE: Final
SOURCE: Scopus
Kyriakopoulos, G.L.
6603382498;
Should low carbon energy technologies be envisaged in the context of sustainable energy systems?
(2021) Low Carbon Energy Technologies in Sustainable Energy Systems, pp. 357-389. Cited 4 times.
https://www.scopus.com/inward/record.uri?eid=2-s2.0-85109526422&doi=10.1016%2fB978-0-12-822897-5.00015-8&partnerID=40&md5=7f4ea6cfcd857b70829a9f6b6fce89bf
DOI: 10.1016/B978-0-12-822897-5.00015-8
AFFILIATIONS: School of Electrical and Computer Engineering, Electric Power Division, Photometry Laboratory, National Technical University of Athens, 9 Heroon Polytechniou Street, Athens, 15780, Greece
ABSTRACT: Nowadays energy demand and supply are facing a steady pressure due to the ongoing growth of global population and the social pursue of humans well being. In response to this, multifaceted socio-economic and environmental phenomenon, at this concluding chapter the main contributors of energy systems of low carbon orientation should proven feasible tools to shape strategic policies of energy planning. These tools were classified into two certain dimensions: technological and social. Regarding the technological dimension the key-aspects of investigation are that of: Mining marketplace and power consumption; resource recovery technologies in reducing use of fossil fuels and fossil-based mineral fertilizers; renewable energy sources and energy crops; offshore wind farms using GIS-based multi-criteria decision analysis and analytical hierarchy process. Besides, key aspects of the social dimension are that of: household sector; education; bibliometric analysis on energy, sustainability and climate change; public acceptance of energy systems based on renewables. Subsequently, the technical constraints, the drivers, the barriers, and the developmental prospects of the analysis were determined, while the challenges of energy systems devoted to carbon abatement and the future research orientations, they have been conclusively outlined. © 2021 Elsevier Inc. All rights reserved.
AUTHOR KEYWORDS: Carbon abatement; Environmental sensitivity; Low carbon technologies; Multi-criteria decision analysis; Public acceptance; Renewable energy sources; Renewable energy technologies; Social participation; Sustainable energy systems
DOCUMENT TYPE: Book Chapter
PUBLICATION STAGE: Final
SOURCE: Scopus
Liu, X., Wang, Q.-C., Jian, I.Y., Chi, H.-L., Yang, D., Chan, E.H.-W.
57214856737;57202906291;57214594997;35096047900;37092552700;57204500597;
Are you an energy saver at home? The personality insights of household energy conservation behaviors based on theory of planned behavior
(2021) Resources, Conservation and Recycling, 174, art. no. 105823, . Cited 5 times.
https://www.scopus.com/inward/record.uri?eid=2-s2.0-85113218051&doi=10.1016%2fj.resconrec.2021.105823&partnerID=40&md5=9c1559d2d887f9c6c9cafc7940dfc146
DOI: 10.1016/j.resconrec.2021.105823
AFFILIATIONS: Department of the Built Environment, Eindhoven University of Technology, Eindhoven, 5600MB, Netherlands;
Department of Land Economy, University of Cambridge, Cambridge, CB3 9EP, United Kingdom;
School of Design, The Hong Kong Polytechnic UniversityHong Kong 999077, China;
Department of Building and Real Estate, The Hong Kong Polytechnic UniversityHong Kong 999077, China
ABSTRACT: Personality traits play an important role in pro-environmental behavioral heterogeneity. This study aims to explore the effects of Big Five personality traits on the energy-saving behavior of residents based on the extended theory of planned behavior (TPB). We employ the k-means algorithm to cluster 1119 respondents in Xi'an, China by their personality characteristics into four groups: (1) the positives, (2) the temperates, (3) the conservatives, and (4) the introverts. The research observes significant heterogeneity of energy-saving behavior among the four resident groups. We examine the behavioral pattern of each resident group, and the analysis indicates that TPB attributes bridge personality traits and household energy-saving behaviors. The extended TPB factors explain the best performance on household energy-saving intention and behaviors of the positives. Besides, the results present the different effects of psychological factors on the energy-saving behaviors of different resident groups. The positive and temperate groups’ energy-saving intention is more sensitive to subjective norms, while perceived behavior control plays a more critical role in other groups. This study could broaden the scope of pro-environmental behavior research and advance knowledge by untangling the intertwined relationship between personality traits and household energy-saving behavior. The findings can contribute to occupant typology development, which is important to capture the diverse energy effect of occupant activity in building energy simulation research as well as differential energy-saving intervention setting in residential buildings to achieve sustainable development goals. © 2021
AUTHOR KEYWORDS: Building energy modeling; Energy-saving behavior; Five factor model; Personality traits; Pro-environmental behavior; Theory of planned behavior
DOCUMENT TYPE: Article
PUBLICATION STAGE: Final
SOURCE: Scopus
Ogbuanya, T.C., Nungse, N.I.
57194210662;57249272300;
Effectiveness of energy conservation awareness package on energy conservation behaviors of off-campus students in Nigerian universities
(2021) Energy Exploration and Exploitation, 39 (5), pp. 1415-1428.
https://www.scopus.com/inward/record.uri?eid=2-s2.0-85114420949&doi=10.1177%2f0144598720975133&partnerID=40&md5=7cd944dc470f7790cb331bd899e87541
DOI: 10.1177/0144598720975133
AFFILIATIONS: Industrial Technical Education, University of Nigeria, Enugu, Nigeria
ABSTRACT: Poor energy conservation behaviors among off-campus students are a form of irrational behavior that often results in erratic power supply within the students' village. The present study, therefore, explored the effectiveness of energy conservation awareness package (ECAP) on energy conservation behaviors of off-campus students residing within students' villages of Nigerian Universities. A pretest-posttest experimental and waitlist control group which involved 328 participants were quantitatively assessed. Findings of the posttest depict that poor energy conservation behavior of off-campus students momentously weaken compared to a waitlist control group. This is evident where the mean difference of 33.35 on level of engaging in energy conservation behavior for students exposed to ECAP is greater than 1.20 for those not exposed to the therapy. Also, 27.97 mean difference on likelihood of engaging in energy conservation behavior is greater than 0.54 for those not exposed to the therapy. In extension, results of 2 and 4-months follow-up appraisal proved that the momentous decline in negative energy conservation behavior of ECAP participants was upheld. By implication, when off-campus students subscribe to the use of energy-saving bulbs instead of incandescent bulbs and switching off the unnecessary lightings among others as evident in this study would go a long way in conserving energy. © The Author(s) 2020.
AUTHOR KEYWORDS: energy consciousness; Energy conservation Awareness Package; Negative energy conservation behavior; off-campus students; students’ village
DOCUMENT TYPE: Article
PUBLICATION STAGE: Final
SOURCE: Scopus
Xu, X., Xiao, B., Li, C.Z.
57188580287;57214779912;57188592789;
Analysis of critical factors and their interactions influencing individual's energy conservation behavior in the workplace: A case study in China
(2021) Journal of Cleaner Production, 286, art. no. 124955, . Cited 8 times.
https://www.scopus.com/inward/record.uri?eid=2-s2.0-85095955705&doi=10.1016%2fj.jclepro.2020.124955&partnerID=40&md5=9385146a37e387394ede3e7eacf36148
DOI: 10.1016/j.jclepro.2020.124955
AFFILIATIONS: School of Civil Engineering, Nanjing Forestry University, Nanjing, China;
Faculty of Social Science, Lingnan UniversityHong Kong, Hong Kong;
College of Civil and Transportation Engineering, Shenzhen University, Shenzhen, China
ABSTRACT: Energy conservation in the building sector is a major target worldwide, and individual's energy conservation behavior is considered a critical means to meet energy conservation goals. Although individual's energy conservation behavior has been explored, its formation mechanism in the workplace remains unclear. This study aims to investigate the effect of the determinants of energy conservation behavior in the workplace based on the extended theory of planned behavior model. The novelty of this study is that it extends the theory of planned behavior with the consideration of actual behavior controls. This study is also one of the first studies to identify the effect of user experience on energy saving behavior. Questionnaire survey was used for data collection, and structural equation modeling was applied to develop the structural model. Results show that individual's comfort is the most critical impact factor, followed by user experience and supervision. Intention has a limited effect on energy saving behavior. Moreover, three strategies were proposed to promote energy conservation behavior in the workplace from the perspectives of design, supervision, and punishment. These findings will not only help researchers in conducting further in-depth research on the mechanism of individual's energy conservation behavior but also practitioners in proposing other effective strategies to promote energy conservation behavior. © 2020 Elsevier Ltd
AUTHOR KEYWORDS: Energy conservation behavior; Individual's comfort; Supervision; User experience
DOCUMENT TYPE: Article
PUBLICATION STAGE: Final
SOURCE: Scopus
Alomari, M.M., El-Kanj, H., Topal, A.
26031239300;57215535881;57202337104;
Analysis of energy conservation behavior at the kuwaiti academic buildings
(2021) International Journal of Energy Economics and Policy, 11 (1), pp. 219-232.
https://www.scopus.com/inward/record.uri?eid=2-s2.0-85097011745&doi=10.32479%2fijeep.10407&partnerID=40&md5=75db09eeba411f91dcc61d2971da1160
DOI: 10.32479/ijeep.10407
AFFILIATIONS: Department of Electrical Engineering, Australian College of Kuwait, Safat, 13015, Kuwait;
Faculty of Economics and Administrative Sciences, Nigde Omer Halisdemir University, Nigde, 51240, Turkey
ABSTRACT: Understanding user’s behavior in buildings is crucial since user behavior significantly contributes to the overall building’s energy consumption. Therefore, this study aims to identify a user’s pro-environmental behavior, in particular, the energy conservation behavior (ECB) of university users in Kuwait. For this reason, this study creates a model whereby two variables, namely, environmental knowledge and awareness of consequences, are introduced and incorporated into the existing theory of planned behavior (TPB). The research data is acquired through questionnaires in keeping with Kuwait’s social norms and culture. The extended TPB model is tested using numerical analysis problems in partial least square structural equation modeling (PLS-SEM) to investigate the following variables: energy conservation behavior, intention, subjective norm, attitude, perceived behavioral control, environmental knowledge, and awareness. Results show the indirect effects of the two above mentioned variables on conservation behavior. The results also reveal that societal pressure and cultures significantly affect the users’ intention to engage in energy conservation behavior. The outcomes of this research suggest that there is a need to encourage energy conservation behavior changes in Kuwaiti academics’ buildings by supporting the antecedents, as well as eliminating barriers to pro-environmental actions. © 2021, Econjournals. All rights reserved.
AUTHOR KEYWORDS: Energy Conservation; Environmental Behavior; Higher Educational Institutes; Partial Least Square Structural Equation Modelling; Theory of Planned Behavior
DOCUMENT TYPE: Article
PUBLICATION STAGE: Final
SOURCE: Scopus
Stankuniene, G., Streimikiene, D., Kyriakopoulos, G.L.
57206727568;57195415199;6603382498;
Systematic literature review on behavioral barriers of climate change mitigation in households
(2020) Sustainability (Switzerland), 12 (18), art. no. 7369, . Cited 16 times.
https://www.scopus.com/inward/record.uri?eid=2-s2.0-85091603148&doi=10.3390%2fSU12187369&partnerID=40&md5=1a7076b8247f73415950cfc01e04e210
DOI: 10.3390/SU12187369
AFFILIATIONS: Laboratory of Energy Systems Research, Lithuanian Energy Institute, Breslaujos str. 3, Kaunas, 44403, Lithuania;
School of Electrical and Computer Engineering, Electric Power Division, Photometry Laboratory, National Technical University of Athens, 9 Heroon Polytechniou Street, Athens, 15780, Greece
ABSTRACT: Achieving climate change mitigation goals requires the mobilization of all levels of society. The potential for reducing greenhouse gas (GHG) emissions from households has not yet been fully realized. Given the complex climate change situation around the world, the importance of behavioral economic insights is already understood. Changing household behavior in mitigating climate change is seen as an inexpensive and rapid intervention measure. In this paper, we review barriers of changing household behavior and systematize policies and measures that could help to overcome these barriers. A systematic literature review provided in this paper allows to define future research pathways and could be important for policy-makers to develop measures to help households contribute to climate change mitigation. © 2020 by the authors.
AUTHOR KEYWORDS: Behavior change; Behavior change barrier; Climate change mitigation; Energy consumption; Households; Nudge and Boost intervention
DOCUMENT TYPE: Review
PUBLICATION STAGE: Final
SOURCE: Scopus
Wang, L., Watanabe, T.
56304459800;55906925100;
Does haze drive pro-environmental and energy conservation behaviors? Evidence from the beijing-tianjin-hebei area in china
(2020) Sustainability (Switzerland), 12 (23), art. no. 9972, pp. 1-18.
https://www.scopus.com/inward/record.uri?eid=2-s2.0-85096878521&doi=10.3390%2fsu12239972&partnerID=40&md5=7dc796bf92f0751ba22587f6e78eb594
DOI: 10.3390/su12239972
AFFILIATIONS: School of Regional Design, Utsunomiya University, 7-1-2 Yoto, Tochigi, Utsunomiya City, 321-0904, Japan;
School of Economics and Management, Kochi University of Technology, 2-22 Eikokuji, Kochi, Kochi City, 782-0003, Japan
ABSTRACT: Humans conduct themselves in relation to energy use; energy use has degraded air quality, as reflected by haze occurrence in countries such as China. Improving the population’s involvement in environmental and energy conservation necessitates understanding their motivation to behave under haze. Considering the social problems caused by haze conditions in China, this study used people’s risk perception as a basis to determine their motivations to perform pro-environmental and energy-saving behaviors. We analyzed motivation from privately and publicly oriented perspectives as well as adaptive and mitigative behavioral viewpoints. Motivation-related data were collected through face-to-face discussion and a survey of 506 respondents in the Beijing-Tianjin-Hebei area, which is one of the most heavily polluted regions in China. We conducted multiple regression analysis to determine the extent to which socio-demographic characteristics and risk perception concerning haze predict motivation and actual behavior. Results showed that these factors explain 36.8% and 30.5% of privately and publicly oriented motivations, respectively, but more strongly explain more adaptive (i.e., privately oriented; 55.0%) than mitigating (i.e., publicly oriented; 8.8%) behaviors. Although the residents are motivated to behave equally for private and public purposes in initial conservation efforts, they tend to exhibit adaptive behavior more frequently than mitigating behaviors. These results serve as a reference in encouraging China’s residents to act pro-environmentally and use energy conservatively, thereby contributing to environmental and energy saving education for the society. © 2020, MDPI AG. All rights reserved.
AUTHOR KEYWORDS: Behaviors; Energy-saving; Haze conditions; Motivation; Pro-environment; Risk perception
DOCUMENT TYPE: Article
PUBLICATION STAGE: Final
SOURCE: Scopus
Rizzi, F., Annunziata, E., Contini, M., Frey, M.
38562153500;55368320300;57203306498;34871770800;
On the effect of exposure to information and self-benefit appeals on consumer's intention to perform pro-environmental behaviours: A focus on energy conservation behaviours
(2020) Journal of Cleaner Production, 270, art. no. 122039, . Cited 12 times.
https://www.scopus.com/inward/record.uri?eid=2-s2.0-85086082511&doi=10.1016%2fj.jclepro.2020.122039&partnerID=40&md5=079a94cf6fb9fc854ce56c723ad3d1a5
DOI: 10.1016/j.jclepro.2020.122039
AFFILIATIONS: Institute of Management, Sant'Anna School of Advanced Studies, Pisa, Italy;
Department of Economics, University of Perugia, Perugia, Italy
ABSTRACT: Technology providers have clearly understood that considering consumers’ behavioural changes is important in developing new technologies, particularly in the energy sector. This study examines the effectiveness of exposure to information and self-benefit appeals in determining the energy conservation behaviours of consumers. In particular, based on an extension of the Theory of Planned Behaviour, we used data from a survey of 450 householders in Tuscany (Italy) to analyse how advertising appeals and the prior exposure to general information about energy conservation influence intentions to undertake energy saving behaviour and invest in different energy efficient technologies. The results suggest that advertising based on self-benefit appeals, which is a communication method typically aimed at producing short-term effects, is effective in promoting reductions in energy consumption and investment in widely-adopted technologies, but cannot increase the interest of consumers in scarcely-adopted ones, which have less associations with repeated exposure to general information about energy conservation. Thus, technology providers should consider combining communication methods with short-term and long-term orientations to successfully turn consumers’ informational basis and self-benefit appeals into intentions to perform pro-environmental behaviours. The study concludes with a discussion of its theoretical and managerial implications in the field of market-oriented technology management. © 2020 Elsevier Ltd
AUTHOR KEYWORDS: Advertising strategies; Communication strategies; Consumer behaviour; Energy conservation behaviour; Theory of planned behaviour
DOCUMENT TYPE: Article
PUBLICATION STAGE: Final
SOURCE: Scopus
Dursun, Ä°., Tümer Kabadayı, E., TuÄŸer, A.T.
54388844500;6503986736;57207980990;
Overcoming the psychological barriers to energy conservation behaviour: The influence of objective and subjective environmental knowledge
(2019) International Journal of Consumer Studies, 43 (4), pp. 402-416. Cited 16 times.
https://www.scopus.com/inward/record.uri?eid=2-s2.0-85063535980&doi=10.1111%2fijcs.12519&partnerID=40&md5=7774323a836b656d416a9bb6a0e5b355
DOI: 10.1111/ijcs.12519
AFFILIATIONS: Faculty of Business Administration, Gebze Technical University, Gebze-Kocaeli, Turkey;
Vocational School, Istanbul Okan University, Istanbul, Turkey
ABSTRACT: Energy conservation is a crucial aspect of responsible consumption which is the reflection of individual efforts for sustainability. However, especially young consumers are reluctant to reduce their energy consumption despite their pro-environmental attitudes. Resistance to behavioural change can be attributed to various psychological barriers that help consumer to avoid engaging in pro-environmental actions. In this context, the first aim of the study is to extend the theoretical and empirical evidence regarding impeding effects of psychological barriers on individual energy conservation behaviour. Secondly, the study investigates the alleviating role of environmental knowledge on those barriers that limit energy conservation. Proposed impeding effects of objective and subjective environmental knowledge on various denial mechanisms, which are in turn expected to hinder energy conservation, were tested using the survey data collected from young Turkish consumers. Results suggest that denial mechanisms hinder young consumers’ energy conservation behaviour indirectly through diminishing feelings of moral obligations. Moreover, it was found that objective environmental knowledge's effect can be used to break down the psychological barriers and to facilitate the change towards more sustainable energy consumption patterns. Implications of the findings and directions for future research are discussed. © 2019 John Wiley & Sons Ltd
AUTHOR KEYWORDS: denial mechanisms; energy conservation; environmental knowledge; personal norms
DOCUMENT TYPE: Article
PUBLICATION STAGE: Final
SOURCE: Scopus
Shi, D., Wang, L., Wang, Z.
35070389800;57205332968;57205333748;
What affects individual energy conservation behavior: Personal habits, external conditions or values? An empirical study based on a survey of college students
(2019) Energy Policy, 128, pp. 150-161. Cited 21 times.
https://www.scopus.com/inward/record.uri?eid=2-s2.0-85059592911&doi=10.1016%2fj.enpol.2018.12.061&partnerID=40&md5=ac8968a5cef9da48c414cc9e526a2ce4
DOI: 10.1016/j.enpol.2018.12.061
AFFILIATIONS: Institute of Industrial Economics, CASS, No. 2 Yuetan Beixiaojie Street, Beijing, 100836, China;
National Academy of Economic Strategy, CASS, No. 28, Shuguangxili Chaoyang District, Beijing, 100028, China
ABSTRACT: It is important to encourage people to form energy conservation habits to increase energy efficiency. The application of social psychology research paradigm in studying energy conservation behavior sheds more light on what conditions are necessary for sustained energy conservation behavior. Based on a survey of 234 college students in Beijing, this study was carried out using the VBN model as its analysis framework and a structural equation model while focusing on whether egocentric values necessarily lead to non-energy conservation behavior and whether altruistic and biospheric values inevitably lead to energy conservation behavior among college students. The following conclusions can be drawn. First, the study partially verified the basic conclusion of the VBN model, that is, values have a significant effect on energy conservation beliefs, which in turn significantly affect personal energy conservation norms. Second, energy conservation beliefs formed by altruistic and biospheric values are translated into real energy conservation norms. However, egocentric values do not significantly affect the attribution of energy conservation responsibility. Moreover, personal energy conservation norms do not translate into energy conservation behavior. Third, individual behavioral habits and external conditions do not promote the translation of personal norms into real energy conservation behavior. © 2019 Elsevier Ltd
AUTHOR KEYWORDS: Energy conservation behavior; SEM; Values
DOCUMENT TYPE: Article
PUBLICATION STAGE: Final
SOURCE: Scopus
Yue, T., Long, R., Liu, J., Liu, H., Chen, H.
45662198200;8396729800;57202715578;57207827835;57050551500;
Empirical study on households’ energy-conservation behavior of jiangsu province in China: The role of policies and behavior results
(2019) International Journal of Environmental Research and Public Health, 16 (6), art. no. 939, . Cited 10 times.
https://www.scopus.com/inward/record.uri?eid=2-s2.0-85062998344&doi=10.3390%2fijerph16060939&partnerID=40&md5=00e351386e5cac41927838ad8e815791
DOI: 10.3390/ijerph16060939
AFFILIATIONS: School of Management, China University of Mining and Technology, Xuzhou, 221116, China
ABSTRACT: With the improvement of living quality and the increase of energy consumption of residents, their energy conservation behavior (ECB) plays an increasingly important role in energy conservation and emission reduction. As a kind of environmental behavior, ECB of residents is a complicated process. In this paper, ECB is divided into four types, considering habit adjustment, quality threshold, efficiency investment, and interpersonal facilitation. A comprehensive conceptual framework is built, adding perception about energy conservation results (PER) and contextual factors from a new perspective. Based on a survey in Jiangsu province of China, this paper examines the impact of intention on behavior under the moderation of contextual factors, as well as the effect of perception of energy-conservation results on intention and ECB by means of multivariate statistical analysis. The results show that the intention of energy conservation is the determinant of behavior, but it does not well transform into behavior, especially into quality threshold and interpersonal facilitation behavior. Different contextual factors have positive effects on the relationship of intention and different behavior. However, modulating effects of contextual factors as amplifiers do not function effectively due to their low rating scores. PER has a positive impact on intention but not on all types of ECB. Finally, this paper presents important implications for policy makers to optimize energy conservation policy. © 2019 by the authors. Licensee MDPI, Basel, Switzerland.
AUTHOR KEYWORDS: Behavior results; Contextual factors; Energy-conservation behavior; Household
DOCUMENT TYPE: Article
PUBLICATION STAGE: Final
SOURCE: Scopus
Wee, S.-C., Choong, W.-W.
57209251819;55902922900;
Gamification: Predicting the effectiveness of variety game design elements to intrinsically motivate users' energy conservation behaviour
(2019) Journal of Environmental Management, 233, pp. 97-106. Cited 32 times.
https://www.scopus.com/inward/record.uri?eid=2-s2.0-85059308090&doi=10.1016%2fj.jenvman.2018.11.127&partnerID=40&md5=195acb0ec02ce3bb66e267c7dc90e196
DOI: 10.1016/j.jenvman.2018.11.127
AFFILIATIONS: Department of Real Estate, Faculty of Built Environment and Surveying, Universiti Teknologi Malaysia, Skudai, Johor Bahru, 81310, Malaysia
ABSTRACT: This research predicted the effectiveness of variety game design elements in enhancing the intrinsic motivation of users on energy conservation behaviour prior to its actual implementation to ensure cost-effective. Face-to-face questionnaire surveys were conducted at the five recognized Malaysian research universities and obtained a total of 1500 valid survey data. The collected data was run with Structural Equation Modeling (SEM) analysis using SmartPLS 3 software. The results predicted the positive effect of gamification on intrinsically motivate the users based on Self-Determination Theory (SDT). The identified nine core game design elements were found to be useful in satisfying users' autonomy, competence and relatedness need satisfactions specified by SDT. This research is useful to guide the campaign organizer in designing a gamified design energy-saving campaign and provide understanding on the causal relationships between game design elements and users' intrinsic motivation to engage on energy conservation. A game-like campaign environment is believed to be created to users by implementing the game design elements in energy-saving campaign, and subsequently users' intrinsic motivation to engage on energy conservation behaviour can be enhanced. © 2018 Elsevier Ltd
AUTHOR KEYWORDS: Energy conservation behaviour; Energy-saving campaign; Game design elements; Gamification; Intrinsic motivation; Self-determination theory
DOCUMENT TYPE: Article
PUBLICATION STAGE: Final
SOURCE: Scopus
Lacroix, K., Gifford, R.
56707219300;7102275161;
Psychological Barriers to Energy Conservation Behavior: The Role of Worldviews and Climate Change Risk Perception
(2018) Environment and Behavior, 50 (7), pp. 749-780. Cited 60 times.
https://www.scopus.com/inward/record.uri?eid=2-s2.0-85040989524&doi=10.1177%2f0013916517715296&partnerID=40&md5=8d9706b07e5bf03cade2ca5976eccc4d
DOI: 10.1177/0013916517715296
AFFILIATIONS: University of VictoriaBC, Canada
ABSTRACT: We proposed and tested a conceptual model of how cultural cognition worldviews, climate change risk perception, and psychological barriers are related to reported energy conservation behavior frequency. Egalitarian and communitarian worldviews were correlated with heightened climate change risk perception, and egalitarian worldviews were correlated with weaker perceived barriers to reported energy conservation behavior. Heightened climate change risk perception was, in turn, associated with fewer perceived barriers to engagement in energy conservation behavior and more reported energy conservation behaviors. The relation between cultural worldviews and perceived barriers was partly mediated by climate change risk perception. Individuals with distinct worldviews perceived psychological barriers differently, and some barrier components were more strongly related to energy conservation behavior than others. Overall, climate change risk perception was the strongest predictor of perceived barriers and of energy conservation behavior frequency. Future efforts should focus on reducing the psychological barriers to energy conservation behavior identified in this study. © 2017, The Author(s) 2017.
AUTHOR KEYWORDS: climate change; cultural cognition theory; energy use; pro-environmental behavior; psychological barriers; risk perception
DOCUMENT TYPE: Article
PUBLICATION STAGE: Final
SOURCE: Scopus
Shen, M., Young, R., Cui, Q.
36701894100;57191488203;7103080112;
The normative feedback approach for energy conservation behavior in the military community
(2016) Energy Policy, 98, pp. 19-32. Cited 19 times.
https://www.scopus.com/inward/record.uri?eid=2-s2.0-84990946975&doi=10.1016%2fj.enpol.2016.08.014&partnerID=40&md5=8b0dfd130d63d71f0798d34393636c0c
DOI: 10.1016/j.enpol.2016.08.014
AFFILIATIONS: Department of Civil and Environmental Engineering, University of Maryland, College Park, MD 20742, United States;
College of Management and Economics, Tianjin University, 92 Weijin Road, Nankai District, Tianjin, 300072, China
ABSTRACT: In the field of energy conservation programs, the behavior-based method, especially the normative feedback approach, has emerged as a cost-effective solution for energy savings. However it remains doubtful whether normative feedback would generate significant energy savings in absence of financial accountability and whether the normative feedback is influenced by the proximity of the comparison groups. Here we test various normative feedback approaches at the Joint Base Andrews in Maryland, with an objective of understanding this approach. We show that the normative feedback approach can lead to 3.4% energy savings, even when residents are not billed for their electricity usage. Through an analysis of covariance, this paper evaluates the effects of different proximities of comparison and concludes that a street-level comparison level can generate the highest energy savings of 5.4%. Furthermore, this paper also explores and defines the relationship between electricity savings and physical variables including home size, unit type, neighborhood, and the variation of cooling degree days. The study contributes to the understanding of how to realizing the full potential of normative feedback approach in energy savings. © 2016 Elsevier Ltd
AUTHOR KEYWORDS: Energy behavior; Energy conservation; Energy efficiency; Normative feedback; Proximity of comparison
DOCUMENT TYPE: Article
PUBLICATION STAGE: Final
SOURCE: Scopus
Sheau-Ting, L., Mohammed, A.H., Weng-Wai, C.
55774635300;55699353500;55902922900;
What is the optimum social marketing mix to market energy conservation behaviour: An empirical study
(2013) Journal of Environmental Management, 131, pp. 196-205. Cited 26 times.
https://www.scopus.com/inward/record.uri?eid=2-s2.0-84886494982&doi=10.1016%2fj.jenvman.2013.10.001&partnerID=40&md5=69f5ada055ee0bc101c87b43fe58a7bc
DOI: 10.1016/j.jenvman.2013.10.001
AFFILIATIONS: Centre for Real Estate Studies, Faculty of Geoinformation and Real Estate, Universiti Teknologi Malaysia, 81310 Skudai., Johor, Malaysia
ABSTRACT: This study attempts to identify the optimum social marketing mix for marketing energy conservation behaviour to students in Malaysian universities. A total of 2000 students from 5 major Malaysian universities were invited to provide their preferred social marketing mix. A choice-based conjoint analysis identified a mix of five social marketing attributes to promote energy conservation behaviour; the mix is comprised of the attributes of Product, Price, Place, Promotion, and Post-purchase Maintenance. Each attribute of the mix is associated with a list of strategies. The Product and Post-purchase Maintenance attributes were identified by students as the highest priority attributes in the social marketing mix for energy conservation behaviour marketing, with shares of 27.12% and 27.02%, respectively. The least preferred attribute in the mix is Promotion, with a share of 11.59%. This study proposes an optimal social marketing mix to university management when making decisions about marketing energy conservation behaviour to students, who are the primary energy consumers in the campus. Additionally, this study will assist university management to efficiently allocate scarce resources in fulfilling its social responsibility and to overcome marketing shortcomings by selecting the right marketing mix. © 2013 Elsevier Ltd.
AUTHOR KEYWORDS: Choice-based conjoint; Energy conservation behaviour; Social marketing; Strategies
DOCUMENT TYPE: Article
PUBLICATION STAGE: Final
SOURCE: Scopus
Delmas, M.A., Fischlein, M., Asensio, O.I.
7006839982;24398905100;55786244800;
Information strategies and energy conservation behavior: A meta-analysis of experimental studies from 1975 to 2012
(2013) Energy Policy, 61, pp. 729-739. Cited 338 times.
https://www.scopus.com/inward/record.uri?eid=2-s2.0-84881660274&doi=10.1016%2fj.enpol.2013.05.109&partnerID=40&md5=3243efe1c34a39207dcbcab491041900
DOI: 10.1016/j.enpol.2013.05.109
AFFILIATIONS: Institute of the Environment and Sustainability, Anderson School of Management, UCLA, La Kretz Hall, Suite 300, Los Angeles, CA 90095-1496, United States;
Institute of the Environment and Sustainability, UCLA, La Kretz Hall, Suite 300, Los Angeles, CA 90095-1496, United States
ABSTRACT: Strategies that provide information about the environmental impact of activities are increasingly seen as effective to encourage conservation behavior. This article offers the most comprehensive meta-analysis of information based energy conservation experiments conducted to date. Based on evidence from 156 published field trials and 525,479 study subjects from 1975 to 2012, we quantify the energy savings from information based strategies. On average, individuals in the experiments reduced their electricity consumption by 7.4%. Our results also show that strategies providing individualized audits and consulting are comparatively more effective for conservation behavior than strategies that provide historical, peer comparison energy feedback. Interestingly, we find that pecuniary feedback and incentives lead to a relative increase in energy usage rather than induce conservation. We also find that the conservation effect diminishes with the rigor of the study, indicating potential methodological issues in the current literature. © 2013 Elsevier Ltd.
AUTHOR KEYWORDS: Energy conservation; Information strategies; Meta-analysis
DOCUMENT TYPE: Article
PUBLICATION STAGE: Final
SOURCE: Scopus
Author Response
Dear Reviewer,
Thank you so much for providing us with an opportunity to revise our manuscript. The following notes detail a re-write that has dealt with your comments. Each comment is presented, followed by the response given to it. Please see the attachment.
We are glad to continue to modify and improve our manuscript if you still have any further questions.
Best,
Xuefeng Chen, Professor
Key Laboratory of Behavioral Science
Institute of Psychology, Chinese Academy of Sciences
4A Datun Road, Chaoyang District, Beijing (100101), P.R. China

Reviewer 4 Report
This is an interesting paper on climate change beliefs and energy-conservation behaviors.
Strengths are the huge sample size (more than 3000 participants) and the analytical approach.
More emphasis could be given on potential differences between males and females, different age groups, different financial background levels, etc.
Are results similar for all those or differential?
It could be elaborated in more detail how current theoretical models of beliefs and decision making are advanced with the present results.
The practical implications could be illustrated in more detailed examples from real-life.
Author Response

(The authors gave the same response as above.)

Round 2
Reviewer 3 Report
At this revised manuscript authors developed a comprehensive analysis regarding the linkages developed among climate change considerations, personality traits among youth, and household energy conservation behaviors reported. Besides, at the revised manuscript authors showed ways of exhibiting energy conservation behaviors and implementing energy-efficiency strategies. Both these outcomes were interrelated since energy conservation can reduce household energy consumption by 10%-30%, while behavior-driven energy conservation strategies required less capital and time investment than other approaches. In this respect the revised manuscript sustains novel features and it can be accepted for publication at the “International Journal of Environmental Research and Public Health” as is.